# Numerical and Physical Modeling of Ponte Liscione (Guardialfiera, Molise) Dam Spillways and Stilling Basin

**Monica Moroni** \*, **Myrta Castellino** and **Paolo De Girolamo**

Dipartimento di Ingegneria Civile Edile e Ambientale (DICEA), Sapienza University of Rome, Via Eudossiana 18, 00184 Rome, Italy

\* Correspondence: monica.moroni@uniroma1.it

**Abstract:** Issues such as the design or reauditing of dams due to the occurrence of extreme events caused by climatic change are mandatory to address to ensure the safety of territories. These topics may be tackled numerically with Computational Fluid Dynamics and experimentally with physical models. This paper describes the 1:60 Froude-scaled numerical model of the Liscione (Guardialfiera, Molise, Italy) dam spillway and the downstream stilling basin. The k-ω SST turbulence model was chosen to close the Reynolds-averaged Navier–Stokes equations (RANS) implemented in the commercial software Ansys Fluent ®. The computation domain was discretized using a grid with hexagonal meshes. Experimental data for model validation were gathered from the 1:60 scale physical model of the Liscione dam spillways and the downstream riverbed of the Biferno river built at the Laboratory of Hydraulic and Maritime Constructions of the Sapienza University of Rome. The model was scaled according to the Froude number and fully developed turbulent flow conditions were reproduced at the model scale (Re > 10,000). From the analysis of the results of both the physical and the numerical models, it is clear that the stilling basin is undersized and therefore insufficient to manage the energy content of the fluid output to the river, with a significant impact on the erodible downstream river bottom in terms of scour depths. Furthermore, the numerical model showed that a less vigorous jet-like flow is obtained by removing one of the sills the dam is supplied with.

**Keywords:** dams; numerical simulations; physical modeling; water management

## 1. Introduction

Water resource management in hydrology involves the processes of planning, developing and managing water resources. Climate change is making these processes more difficult to deal with [1]. Water storage has always represented an essential task for human activity, with significant implications for flood control or the generation of electricity. From this point of view, dams represent a suitable system to divert water, control flooding and produce hydroelectricity.

All these processes are sensitive to the complex three-dimensional flow effects involved in dam hydrodynamics. To accurately study the hydrodynamics and the fluid–structure interaction issues, Computational Fluid Dynamics (CFD) numerical tests together with experimental models are considered within the present research study as mandatory tools (as shown by [2,3]). CFD solves the governing equations of fluid-flow problems, i.e., the continuity, the Navier–Stokes and the energy equations. Because of the nonlinear terms in these equations, analytical methods yield very few solutions. Then, numerical methods, i.e., CFD, are used to obtain the required solutions. Numerical models prescribe the discretization of the domain. The continuous spatial and temporal domain of the problem must be replaced by a discrete one made up of grid points or cells and time levels. The governing equations of the problem must be replaced by a set of algebraic equations with the grid points/cells and the time levels as their domain. Finally, the solutions at each grid point/cell are obtained when advancing from one time level to the

next. Conversely, a physical model consists of the "physical" reproduction of a scaled artifact and the phenomena that occur in it. Experimental tests performed on physical models provide useful information on the entity and behavior of the variables involved in the phenomena under investigation in a controlled environment. In general, those quantities may be measured in a limited number of points within the domain.

The construction of a physical model can be time- and cost-ineffective. Moreover, a physical model may be affected by scale effects, since not all physical conditions present in nature are reproducible at a laboratory scale. This is especially true in the case of turbulent phenomena. Then, in some cases, a numerical model is the only tool to answer questions related, for instance, to the suitability of existing dams to manage discharge increases with respect to the design values or modifications of the dam geometry. In addition, numerical models make it possible to easily evaluate and compare different scenarios. Nevertheless, the physical model, when available, represents an important tool for verifying and calibrating the results provided by numerical models [4].

The remarkable technological advances of recent decades have made it possible to develop increasingly refined numerical models, allowing the study of the temporal evolution of the fluid features with a spatial resolution which can be very high. The authors in [5–8] presented some of the first examples of numerical simulation applied to the reconstruction of flow over a spillway with a 3D Reynolds-averaged Navier–Stokes (RANS) model. The reliability of numerical models in capturing the water surface profile along dam spillways located in different parts of the world is demonstrated in a few contributions [9–15]. Ref. [16] investigated the hydraulic characteristics of the dam discharge flow and its downstream impact by employing Reynolds-averaged Navier–Stokes equations with the RNG k-$\varepsilon$ eddy viscosity model for its turbulence closure, as well as the volume of fluid method. Complex turbulent flow patterns, including collision, reflection and vortices, were captured by three-dimensional simulation.

The published results encourage the use of numerical models for assessing the hydraulic performance of structures. Furthermore, [17] shows how 3D flood numerical simulations can qualitatively and quantitatively assess flood hazards and serve as a visual reference for the development of flood control schemes, providing an important foundation for flood forecasting, dam design and flood control system application.

This paper describes the 1:60 Froude-scaled numerical model of the Liscione dam spillway and the downstream stilling basin. The k-$\omega$ SST turbulence model was chosen to close the Reynolds-averaged Navier–Stokes equations (RANS), due to its remarkable robustness and reliability in simulations involving similar geometries. The Autocad $^{\circledR}$ software was used to construct the geometry of the computational domain, whereas the simulations were performed with the commercial software ANSYS Fluent $^{\circledR}$. The discretization of the domain was performed via the software provided by ANSYS (Fluent Meshing), which guarantees the generation of a simply connected domain (Watertight Geometry). Data for validating the numerical model were gathered by means of a 1:60 scale physical model built at the Laboratory of Hydraulic and Maritime Constructions of the Sapienza University of Rome. The model was scaled according to the Froude number and fully developed turbulent flow conditions were reproduced at the model scale (Re > 10,000) [18]. In [19], the physical model, the experimental campaign conducted to investigate the key hydrodynamical parameters such as hydraulic levels and hydraulic jump location are described in detail. Furthermore, an innovative technical solution suitable to protect the riverbed located just downstream of the stilling basin by means of artificial Antifer blocks is also illustrated.

The Liscione dam was affected between 24 and 25 January 2003 by a serious rainfall event that caused extensive damage. The rain intensity of the event was measured by two weather stations and the related inflow and outflow rates were quantified. The outflow rates turned out to be 830.0 m$^3$/s, which caused the maximum allowed elevation into the reservoir, i.e., 125.5 m a.s.l., to be overcome. The return period was 30 years. The event caused extensive damage both upstream and downstream of the stilling basin: the failure

and breakage of some concrete elements at the end of the dam chute on the right side of the river; damaged and removed gabions on both the left and right banks, and displaced the bottom protection in the central area of the riverbed.

The event demonstrated that the stilling basin of the Liscione dam was ineffective in dissipating the flow energy content with potential severe effects on the stability of the downstream unprotected riverbed due to massive erosion phenomena. To tackle the scour issue, a few measures are available: the redesign of the existing stilling basin, the redesign of the dam as a whole, replacing the dam elements that contribute to the formation of jet-like flows downstream from the chute; and the implementation of protection strategies employing boulders properly arranged in the riverbed downstream of the stilling basin. The aim of this paper was to demonstrate that a validated numerical model, efficiently implemented using a commercial software and a reasonably powerful PC, can be usefully employed for the reconstruction of the hydrodynamics of existing dams which may need either maintenance or upgrading works, such as in the case of flood discharge increments, but also for the design of novel dams [20].

## 2. Materials and Methods

### 2.1. The Liscione Dam

The Liscione dam is located in the municipality of Guardialfiera in Molise (central Italy). Its construction, which took place between 1967 and 1973, had as a main objective the creation of an artificial reservoir, namely Lake Guardialfiera, by collecting water from the Biferno river (Figure 1a).

The reservoir (Figure 1b), obtained with a barrier in loose materials sealed with a bituminous conglomerate coating, was aimed at flood retention and water storage for irrigation purposes. Figure 1c presents a detailed view of the main dam elements, i.e., the surface spillway, chute, stilling basin and bottom outlet.

Important features of both the reservoir (Lake Guardialfiera) and the Liscione dam are listed in Table 1. Characteristic discharge values and return periods of the catchment area of the dam are shown in Table 2.

The dam surface spillway consists of an ungated ogee weir, 92 m long with a crest elevation at 125.5 m a.s.l. (Figure 2a; detailed view in Figure 2b) and a gated weir with three 13 m wide openings, equipped with automatic flap gates with counterweights pivoted at the sill (Figure 2a; detailed view in Figure 2c). The gate configurations are either open (minimum elevation of 122.0 m a.s.l.) or closed, sharing the same elevation as the ogee weir (i.e., 125.5 m a.s.l.). The gate drop takes place automatically and progressively as soon as the reservoir water level reaches an elevation of 125.5 m a.s.l.

Water collected by the surface spillway is conveyed into the stilling basin via a chute. The channel has a uniform rectangular section of 25 m in length and a horizontal development that is 180 m long. If the water stored in Lake Guardialfiera reaches an elevation of 129 m a.s.l., the ungated ogee spillway and the gated spillway release discharge values of 1080 m$^3$/s and 1174 m$^3$/s, respectively.

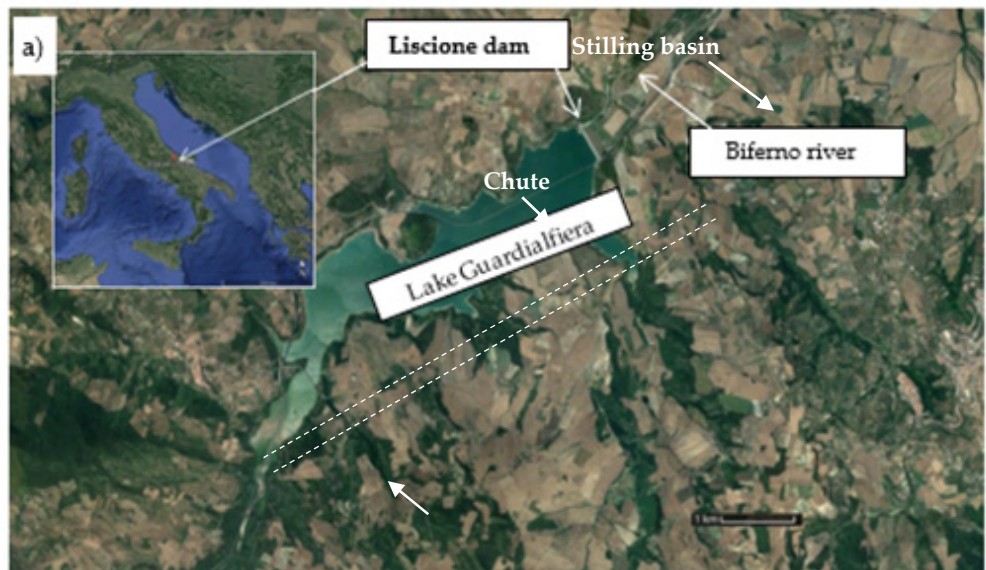

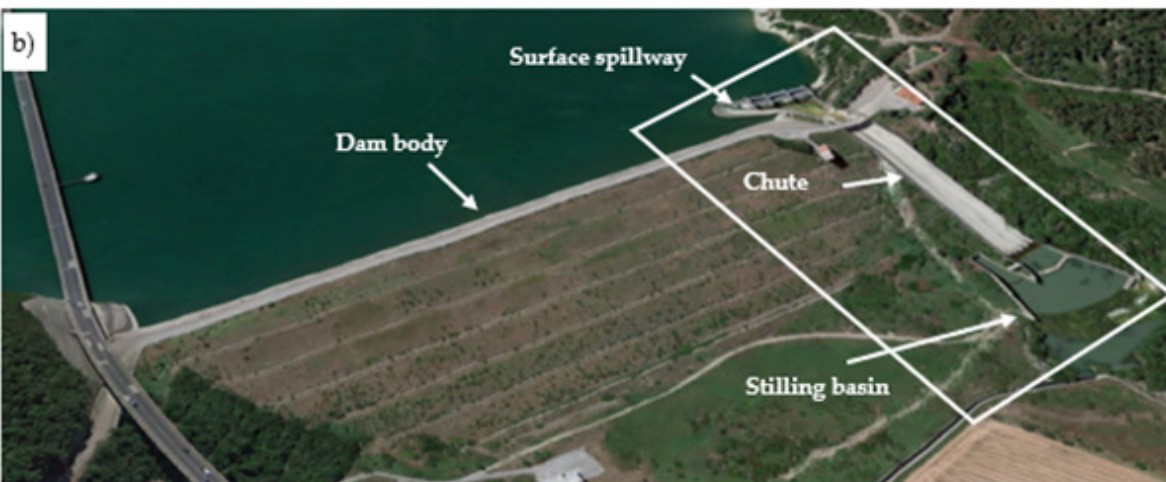

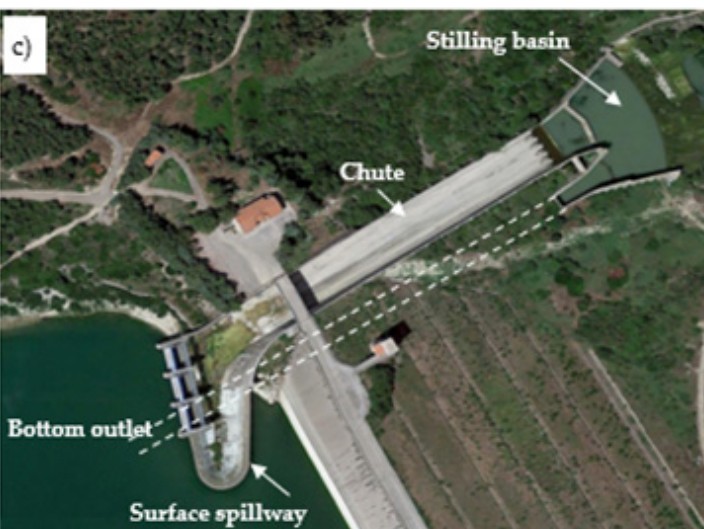

**Figure 1.** (**a**) The investigated area: geographical framework [Map data: Google]; (**b**) overview of the dam body and main elements; (**c**) detailed view of the main elements of the dam, i.e., surface spillway, chute, stilling basin and bottom outlet.

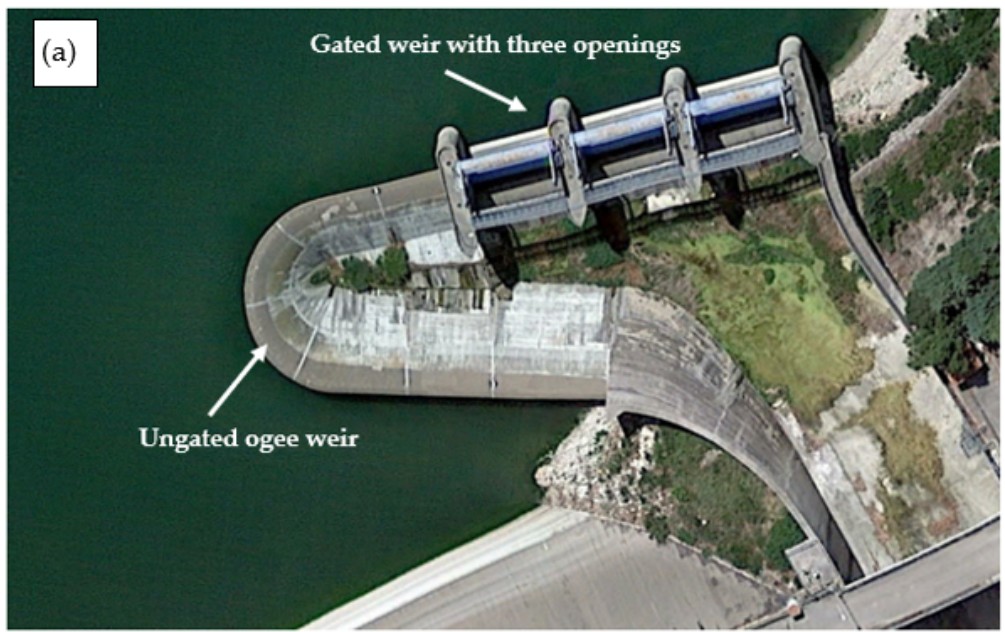

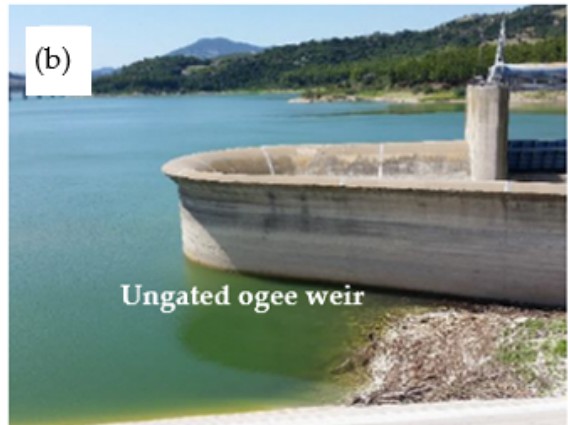

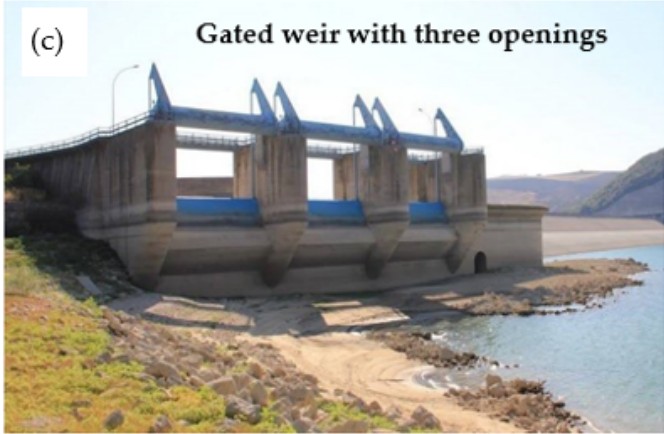

**Figure 2.** Surface spillway: (**a**) top view of the spillway; (**b**) ungated ogee weir (free Creager-type sill); and (**c**) gated weir with three 13 m wide openings, equipped with automatic flap gates.

**Table 1.** Important features of the reservoir (Lake Guardialfiera) and the Liscione dam.

| | Total volume (millions of $m^3$) | Useful storage (millions of $m^3$) | Dead storage capacity (millions of $m^3$) | Reservoir maximum surface ($km^2$) | Surface of the catchment area ($km^2$) |
|---|---|---|---|---|---|
| Reservoir | 173.0 | 137.0 | 11.0 | 7.45 | 1043 |
| | Management upper storage elevation (m a.s.l.) | Maximum allowed water elevation (m a.s.l.) | Dam crest (m a.s.l.) | Management minimum operating level (m a.s.l.) | Minimum foundation height (m a.s.l.) |
| Dam | 125.5 | 129.0 | 131.5 | 92.0 | 71.5 |

**Table 2.** Characteristic discharge values and return periods of the Liscione dam catchment area.

| Discharge inlet to the reservoir ($m^3/s$) | 1050 | 1800 | 2300 | 2650 |
|---|---|---|---|---|
| Discharge at the spillways ($m^3/s$) | 830 | 1450 | 1850 | 2250 |
| Return period (y) | 30 | 200 | 500 | 1000 |

To prevent dam overtopping, a bottom outlet is realized to convey water into the stilling basin (Figure 1c). It consists of a tunnel with an internal diameter of 7.2 m, a length of 309.5 m and a slope of 1%. Its intake is placed at an elevation of 76.4 m a.s.l. and the outlet is at 73.5 m a.s.l. If the water stored in Lake Guardialfiera reaches an elevation of 129 m a.s.l., the bottom outlet is activated to drain a flow rate of approximately 500 m$^3$/s.

To reduce the kinetic energy of water drained into the stilling basin, four nappe splitters are placed at the end of the chute (Figure 3). To the same aim, the stilling basin is equipped with four sills: the first one is placed downstream of the bottom outlet (sill#1), two other sills are placed at the downstream boundary of the chute (sill#2, a sky-jump-like sill, and sill#3) and the last one is at the end of the stilling basin (sill#4). Sill#4 is higher at the hydraulic left of the stilling basin to better dissipate the energy, which in that area, due to the slight curvature of the riverbed, may lead to massive erosion.

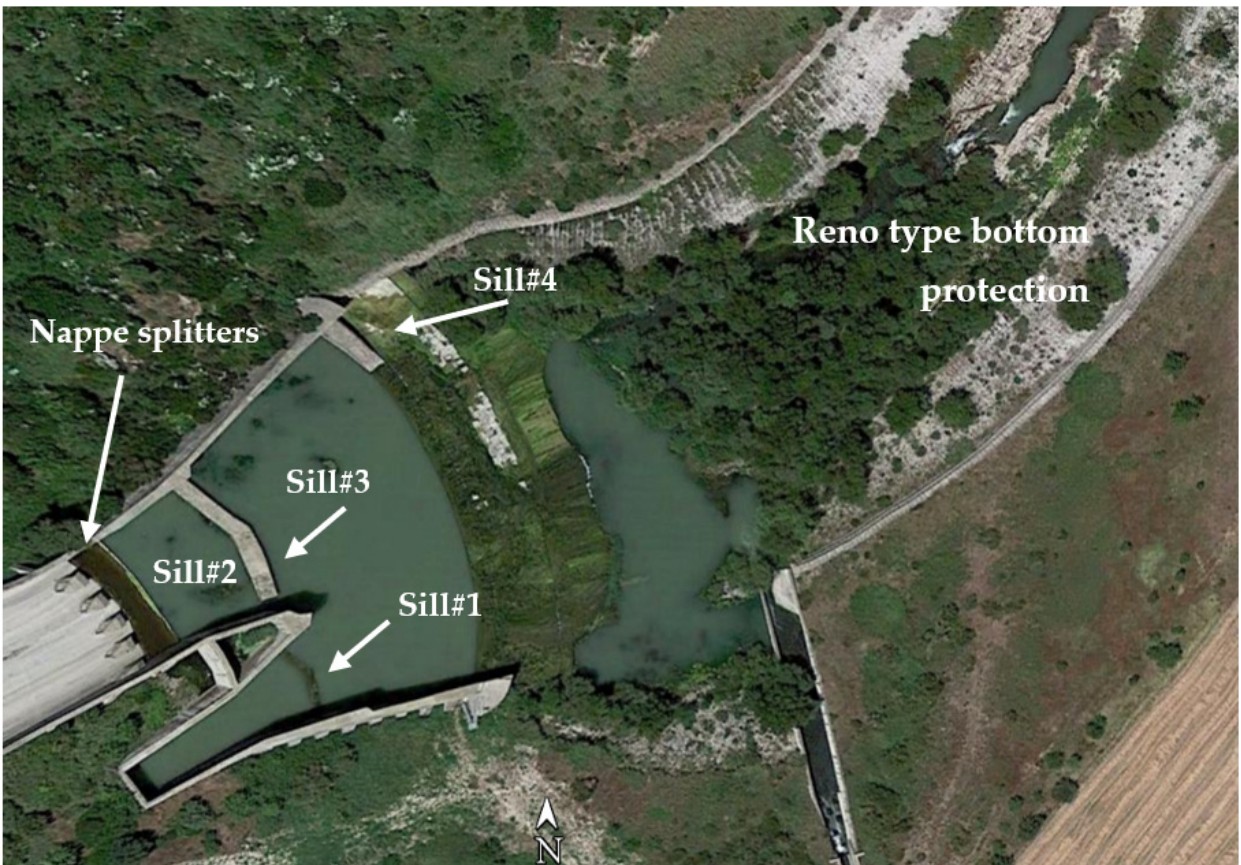

**Figure 3.** Dam elements aimed at reducing the kinetic energy of water drained into the stilling basin.

Downstream of the stilling basin, the central area of the first 500.0 m of the riverbed was covered with 0.3 m thick Reno type bottom protection, and the right and left banks of the riverbed protected by 1.0 m high gabions. After that distance, the riverbed presents the natural waterway.

### 2.2. Experimental Investigation

The physical model was realized in the DICEA-Sapienza University of Rome Hydraulic and Maritime Construction Laboratory. Referring to Figure 4, the physical model was designed in such a way that the following requirements were met:

- The model tank (mimicking the prototype reservoir) dimensions made it possible to include the surface spillway and to ensure a constant water level in the tank up to the maximum tested flow rate, as occurs in reality, due to the large size of the artificial basin. A preliminary investigation demonstrated that a tank with dimensions

shown in Figure 4, i.e., approximately 150 × 210 m, was sufficient to guarantee the above requirement;

- The physical model downstream section was set considering the area wherein the river protection interventions were planned to take place. In addition, the model includes the river portion downstream of the stilling basin characterized by an irregular planimetric geometry. Based on these requirements, the downstream closure section of the model was set after the first river bend, as highlighted in Figure 4.

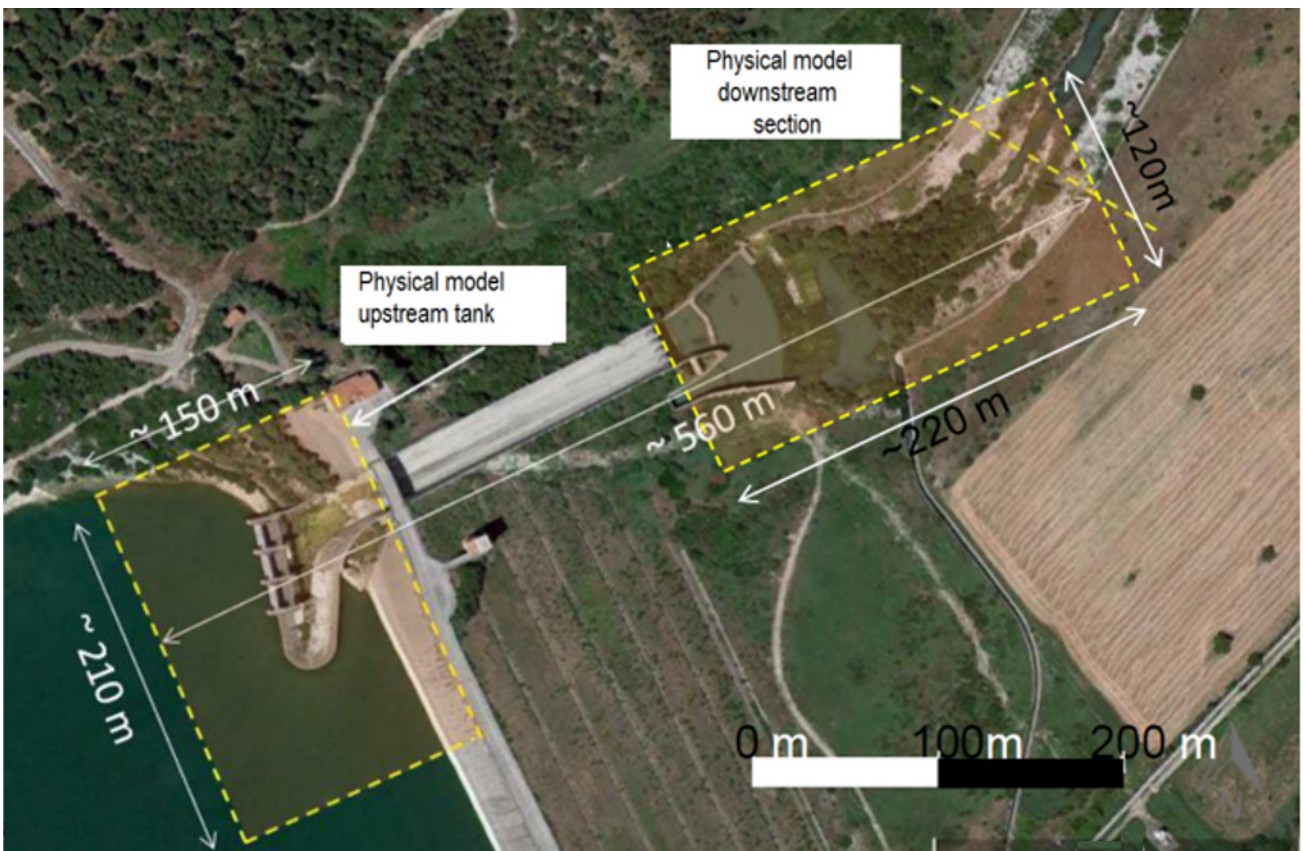

**Figure 4.** Area reproduced with the physical model.

Due the above requirements, the prototype dimensions of the area reproduced with the physical model were 560.0 m as the longitudinal extension and roughly 210.0 m as the maximum width (see Figure 4). A geometric reduction scale of 1:60 results from the adoption of the dimensions reported above.

The components of the spillway, i.e., the three gates (in their lowered configuration), the ogee weir, the chute, the bottom outlet terminus, the sills, and the stilling basin, mimicking the prototype counterparts, are shown in Figure 5. A detailed description of the physical model can be found in [19].

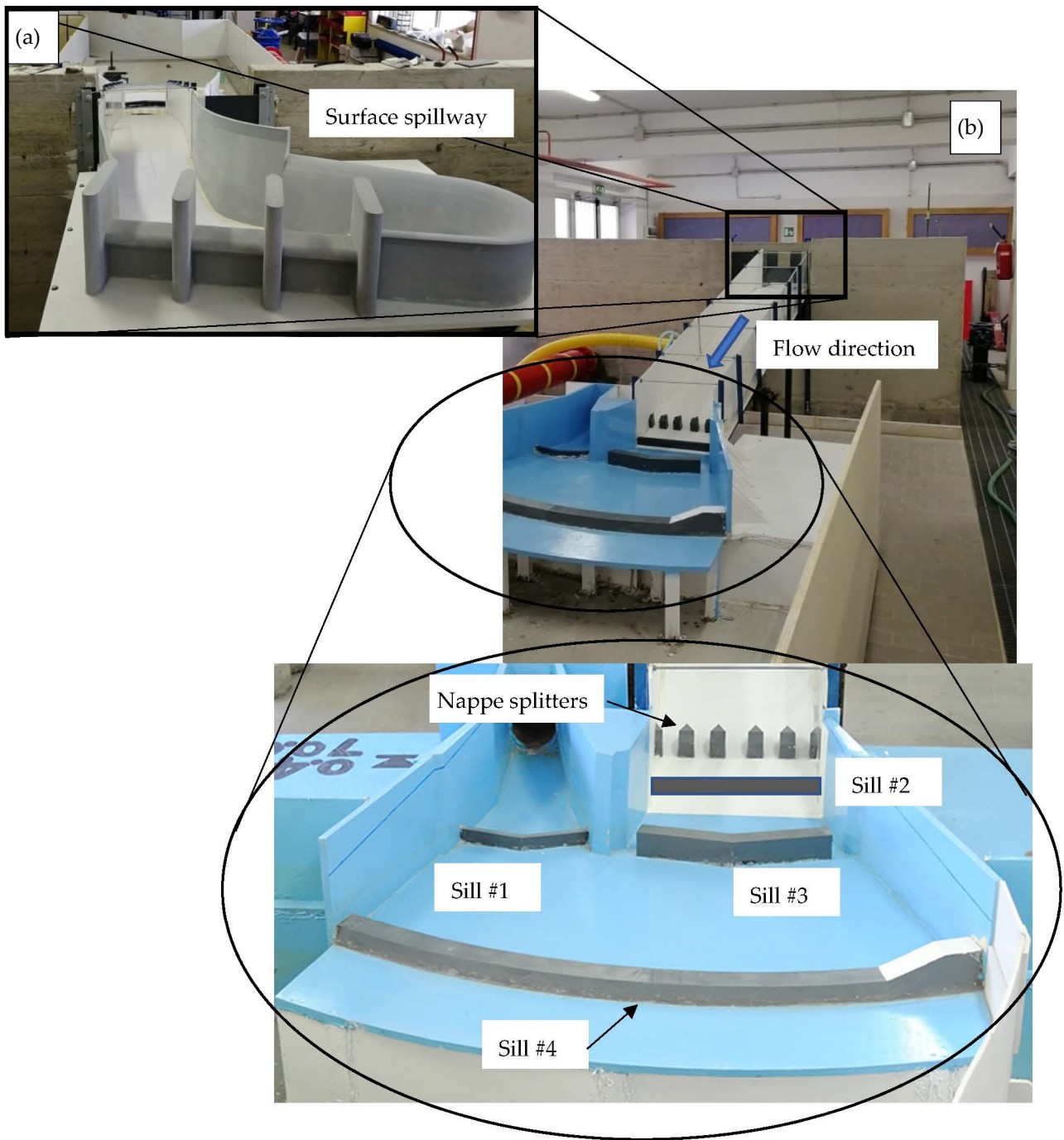

**Figure 5.** Physical mode: (**a**) surface spillway; (**b**) chute, bottom outlet terminus, sills and stilling basin.

### 2.3. Numerical Simulations

A geometry similar to that tested experimentally was investigated with the numerical model. Two sets of numerical simulations were carried out and will be described in the following sections, namely the upstream tank that reproduces the Lake Guardialfiera and the surface spillway (Model #1) and the complete model enclosing the upstream tank, the surface spillway, the chute, the stilling basin, and a small portion of the riverbed downstream of the stilling basin (Model #2). Runoff conditions on the surface spillway were numerically reproduced by imposing a constant water level in the tank. In some preliminary numerical tests, an upstream tank of dimensions larger than those employed for the physical model were tested. No remarkable differences were noticed in terms of the fluid-free surface features and fluid height above the surface spillway. Model #1 outcomes

were employed to determine the stage–discharge rate curve, which was compared to the experimental one. Model #2 outcomes made it possible to investigate the impact area of the jet impinging into and downstream of the stilling basin. Simulations were performed both considering the presence of the ski-jump sill (sill#2) at the foot of the chute and without it to evaluate the distance of impingement of the jet outflowing from the spillway chute in both cases.

To perform the numerical simulations of Model #1 and Model #2, the 3D drawings of the spillway and of the whole dam in the model scale were realized with AutoCAD, starting from the planimetric and cross-sections provided by the dam concessionaire and appropriately compared with the technical drawings realized by the designer. These drawings reported in Figure 6 were imported in Fluent in the Geometry component section of the software.

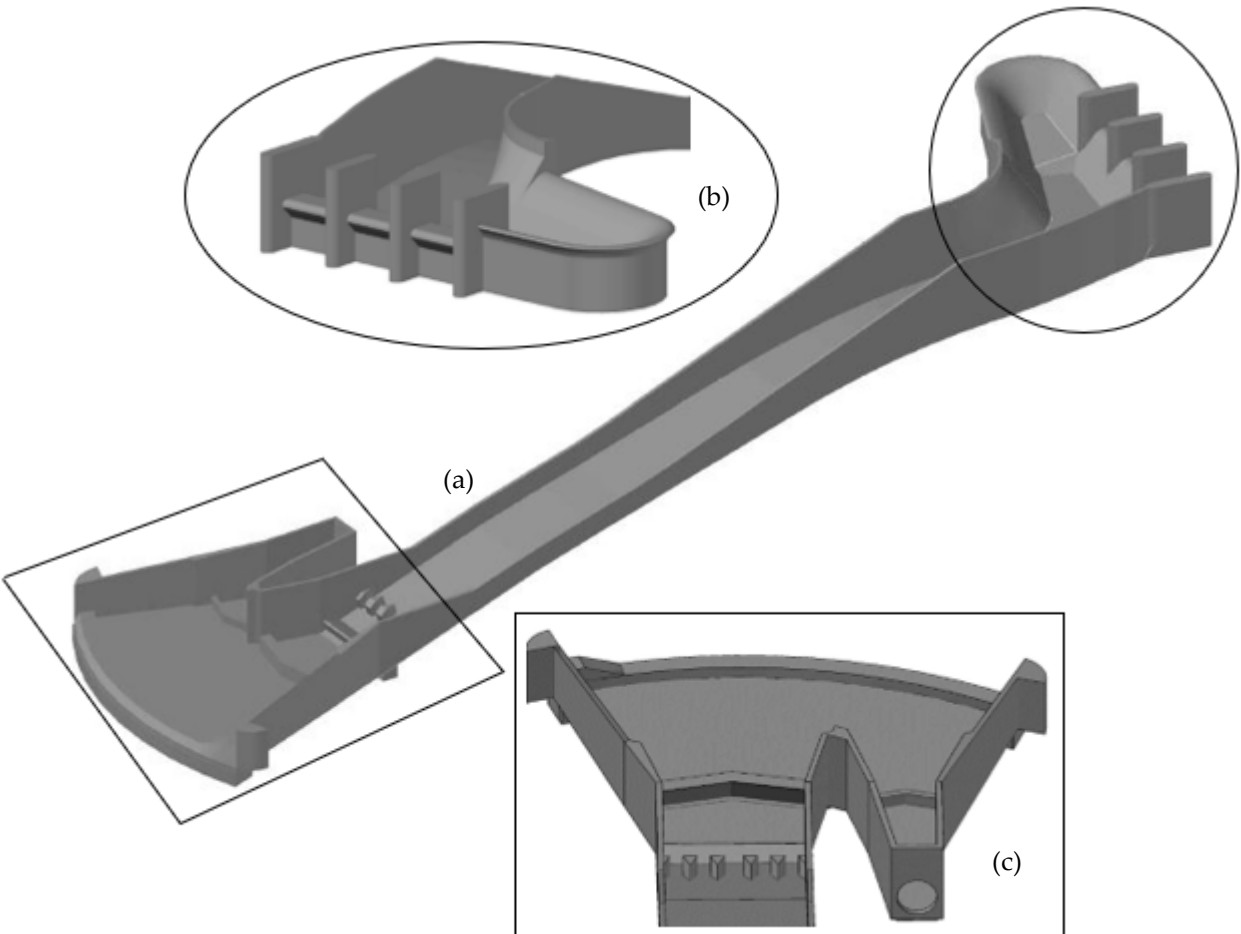

**Figure 6.** (**a**) Three-dimensional drawing of the infrastructure in the model scale carried out with AutoCAD; (**b**) AutoCAD model of the surface spillway; and (**c**) AutoCAD model of the stilling basin.

Initially, the bathymetry was imported as an STL file built from information gathered from the area Digital Terrain Model. Preliminary tests demonstrated that its influence on the reconstructed water levels was negligible with respect to the analogous simulations performed without implementing the lake bottom profile. For this reason, the bathymetry was not included in the final configuration of both Models #1 and #2.

For all models, the computational domain was discretized using a grid with hexagonal meshes. To verify the independence of the results from the mesh size, several simulations were carried out, doubling the number of elements or, when an excessive computational burden was expected, reducing the mesh size by at least 20% in each direction. Table 3

shows the main features of the discretization adopted for the models listed above, specifically the minimum and maximum size of the grids and the total number of elements.

**Table 3.** Main features of the grids employed for both models.

| Model # | Model Description | Resolution | Minimum Size (mm) | Maximum Size (mm) | Number of Elements |
|---|---|---|---|---|---|
| 1 | Surface spillway | Low | 10 | 20 | 259,148 |
| | | High | 5 | 12 | 887,386 |
| 2 | The whole dam | Low | 10 | 20 | 485,258 |
| | | Medium | 8 | 16 | 879,883 |
| | | High | 5 | 12 | 1,024,524 |

The computational domains for the geometries listed above are shown in Figure 7 for Model #1 and Figure 8 for Model #2.

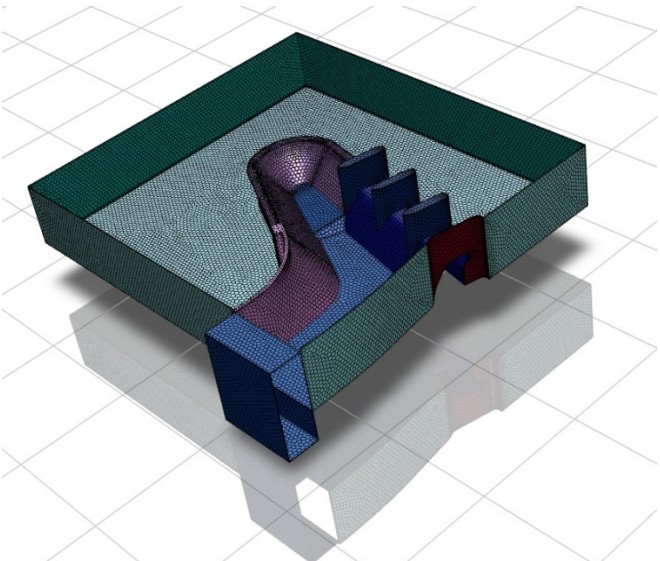

**Figure 7.** Computational domain for Model #1.

For both models, the inlet boundary condition was of the "pressure inlet" type. It was provided by assigning the height of the free surface inside the upstream tank. The "pressure outlet" boundary condition was set at the surfaces in contact with the atmosphere. It was also applied on the walls of the step at the toe of the stilling basin. The step was introduced in the numerical model following [21] since it makes the results more consistent with the experimental evidence. The "wall" boundary condition was assigned to the walls of the dam body including the sills, the chute, and the bottom surface of the stilling basin. The "no slip" condition was set, which prescribes the fluid to adhere to the interface with the wall and moves with the same velocity, and zero velocity in our models since the walls are fixed.

The Fluent software requires an initial value of the water volume fraction (WVF). Inside the upstream tank, the WVF of a certain number of cells was assigned the value 1. Those cells were selected, ensuring a water level slightly above the free surface height provided by the "Inlet" boundary condition. This made it possible to provide initial conditions that were not too far from those of the final solution, considerably reducing the computational cost of the simulations.

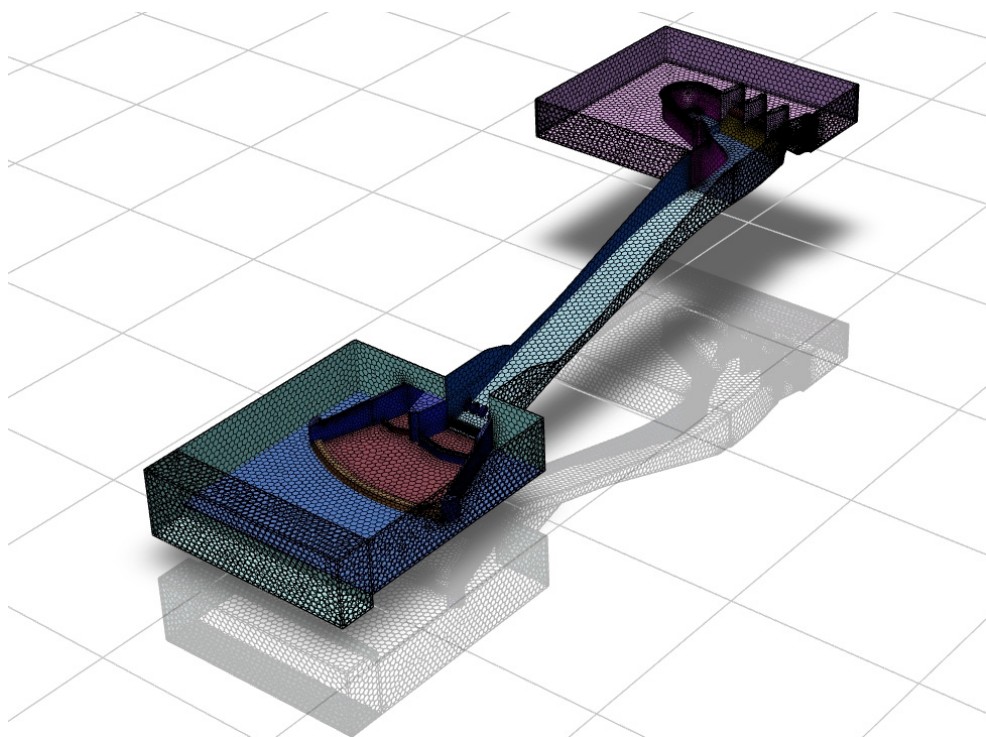

**Figure 8.** Computational domain for Model #2.

Velocity values measured with a Pitot tube were used to validate the numerical models (Figure 9).

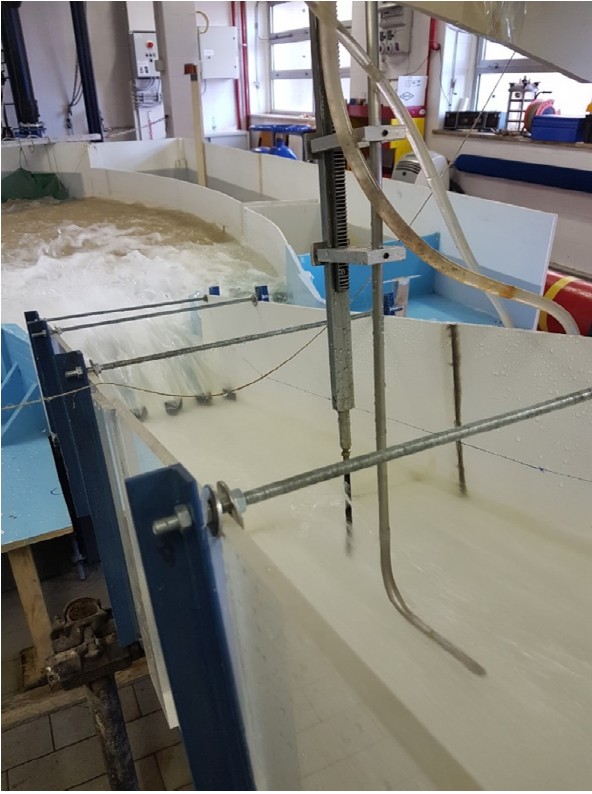

**Figure 9.** Pitot tube placed at (roughly) half the chute length.

These measurements were performed at a discharge rate of 1450 m³/s, i.e., the design flow rate for the hydraulic structure. The velocity profiles in two different points were measured:

(1)    At 0.14 m upstream from the chute, within the surface spillway volume;
(2)    At 1.639 m downstream from the chute (see Figure 9).

## 3. Results

*Model #1: Surface Spillway*

The numerical model including the surface spillway was implemented using two grids of different resolution. For both grids, the simulation was interrupted up to the achievement of the steady-state condition for the flow field.

Table 4 presents the complete set of numerical simulations performed for Model #1, namely the height of the free surface inside the upstream tank, the expected flow rate, the turbulence model adopted and the grid resolution. The expected flow rate for a given height of the free surface was determined from the experimentally achieved stage–discharge rate curve.

**Table 4.** Height of the free surface inside the upstream tank, expected flow rate and turbulence model adopted for Model #1.

| Height of the Free Surface Inside the Upstream Tank (m) | Expected Discharge (m³/s) | Turbulence Model | Grid Resolution |
|---|---|---|---|
| 0.21327 | 304 | k-omega | Low |
| 0.22357 | 424 | k-omega | Low |
| 0.22937 | 530 | k-omega | Low |
| 0.24077 | 830 | k-omega | Low |
| 0.24077 | 830 | k-omega | High |
| 0.25697 | 1450 | k-omega | Low |
| 0.25697 | 1450 | k-omega | High |
| 0.26237 | 1650 | k-omega | Low |
| 0.26237 | 1650 | k-eps | Low |
| 0.26747 | 1850 | k-omega | Low |
| 0.27897 | 2250 | k-omega | Low |

Low discharges were simulated to characterize the initial portion of the stage-discharge rate curve. Q = 830 m³/s was the maximum flow rate discharged from the spillway during the event which occurred in January 2003, characterizing a rainfall event with a return period of 30 years. Q = 1450 m³/s, Q = 1650 m³/s, Q = 1850 m³/s and Q = 2250 m³/s correspond to a return period of 100 years, 200 years, 500 years and 1000 years, respectively.

Figure 10 qualitatively compares the hydrodynamics that occurred during the flood event of January 2003 (Figure 10a), the outcome of the laboratory experiment (Figure 10b) and the numerical model (Figure 10c).

Though the air entrainment at the prototype scale appears more evident, the physical and numerical models present similar flow features.

To determine the best parameters to employ within the simulations, the numerical model for a prototype discharge value equal to 1450.0 m³/s was run for both a low- and a high-grid resolution. The k-omega SST model was used as a turbulent model. The comparison between the hydrodynamics resulting from the numerical model (for the high grid resolution case) and the physical model is presented in Figure 11. The numerical model output of the low resolution case looks very similar to Figure 11a, and for this reason, it was not presented. Moreover, the numerical model outputs are very similar to the

water-free surface provided by the physical model. This suggests that the low-resolution grid was sufficiently refined to describe the phenomenon under investigation. The grid independence of the solution is further demonstrated by analyzing the velocity profile located 0.14 m upstream from the chute, within the surface spillway volume where the Pitot tube measurements were available for the discharge value under investigation. Figure 11c presents the comparison between the velocity values at different heights calculated with the high- and low-resolution models and corresponding values measured with the Pitot tube. No remarkable difference can be noticed between the numerical profiles which appear to be slightly overestimated with respect to the measured velocity values, as was to be expected due to the intrusive nature of the measurement with the Pitot tube, which may affect the magnitude of the velocity value detected. Considering this result, the low resolution grid was employed for all simulations.

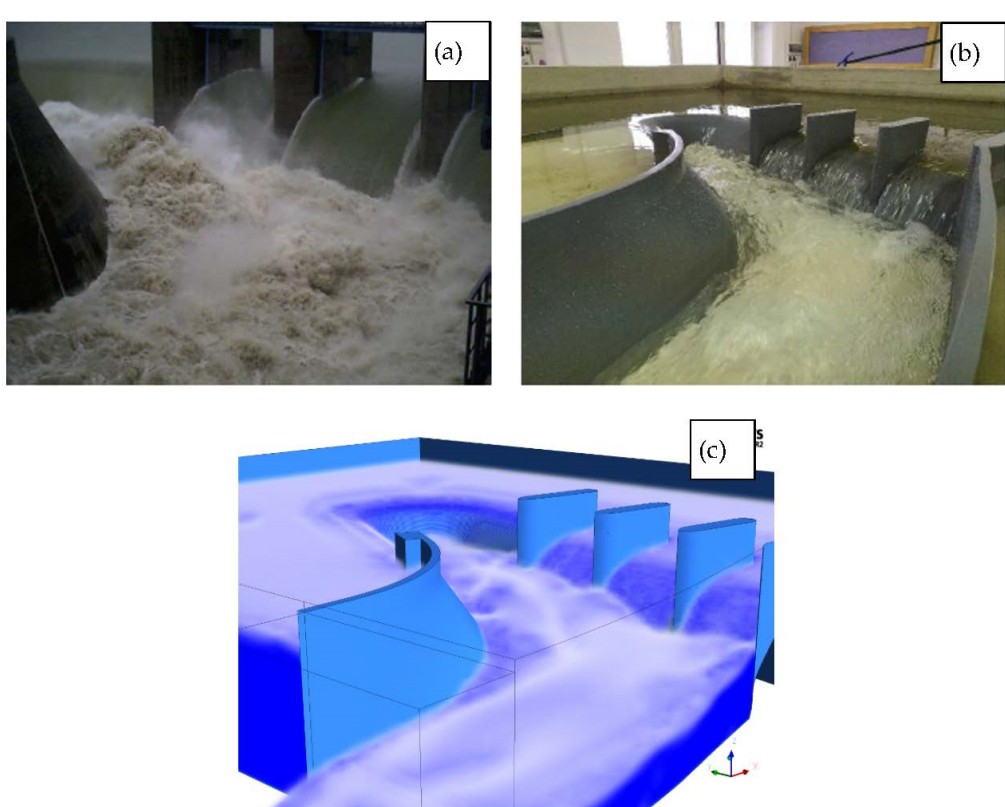

**Figure 10.** Comparison among the 2003 flow event (**a**) with a picture of the real event, (**b**) the physical model output and (**c**) the result of the numerical simulation.

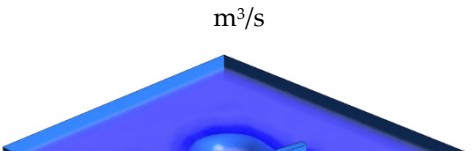

High resolution model—Discharge 1450 m³/s

Physical model—Discharge 1450 m³/s

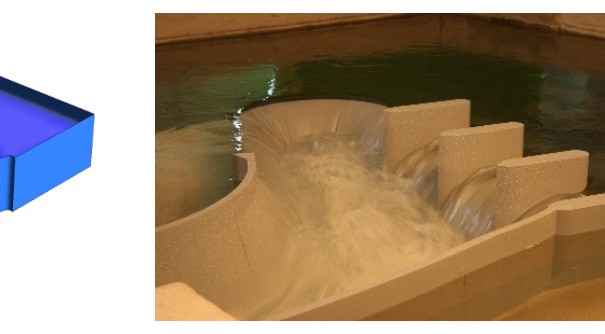

**Figure 11.** *Cont.*

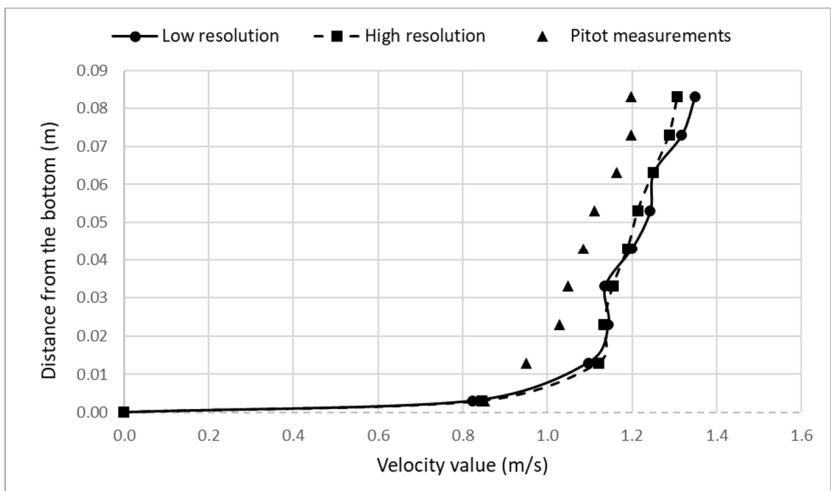

**Figure 11.** Comparison between the hydrodynamics resulting from the numerical model (for both the tested grid resolutions) and the physical model.

A further test was conducted to choose the turbulent model providing the best results in terms of similarity between the numerical output and physical model. The k-omega SST and k-eps models were employed and, as shown in Figure 12, no remarkable differences can be noticed between the output of the numerical model employing the k-omega SST turbulence model and the output of the physical model. Once again, the free surfaces output by the numerical models employing the k-omega SST and the k-eps turbulence models look very similar. For this reason, the k-omega SST model was used for the simulations.

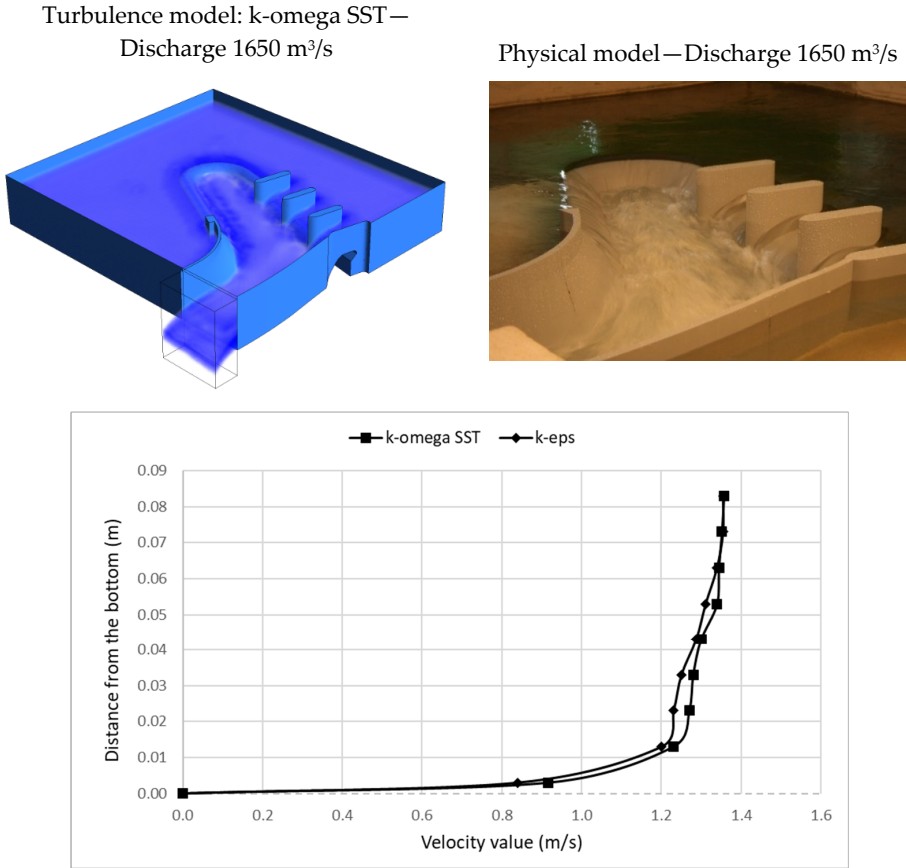

**Figure 12.** Comparison between the hydrodynamics resulting from the numerical model (for both the tested turbulence models) and the physical model.

In order to quantitatively check the reliability of the numerical model, the discharge computed for each model (and the corresponding water stage employed as the inlet boundary condition) was compared to that employed within the experimental investigation. The experimental procedure consisted in varying the discharge flowing through the model and contextually measuring the water level within the upstream tank that reproduces the modeled Lake Guardialfiera. Eighteen experiments were run with different values of the discharge. The experimentally obtained stage–discharge rate curve was then compared to that one used at the dam design stage and validated. The comparison was satisfactory as the maximum error was roughly equal to 3% (refer to [19] for the complete procedure employed to construct the experimental rating curve).

For the numerical model, the discharge was computed assuming a threshold value for the simulated water volume fraction and computing the flow rate value as the product of the average velocity and the area of the water section. Different threshold values were considered, ranging from 0.3 to 0.7 with step 0.1, and the value 0.6 appeared to be the one providing the best agreement between the experimentally detected rating curve and the numerical one. The two curves are displayed in Figure 13.

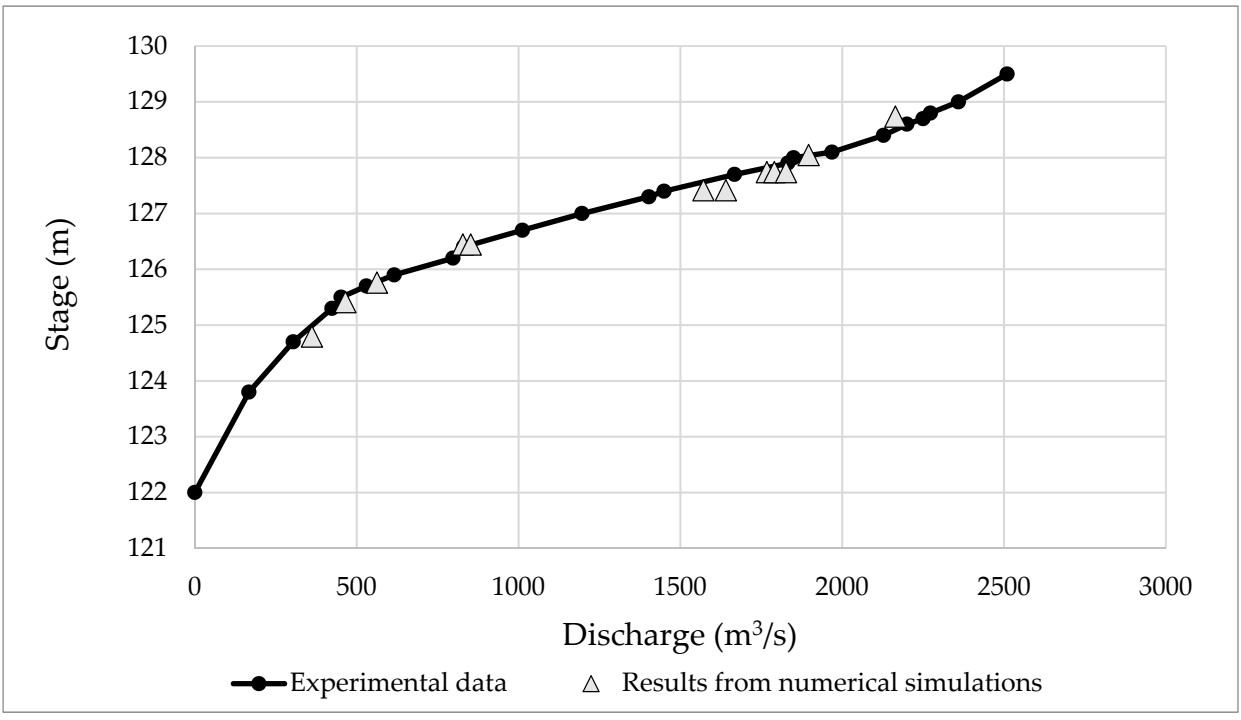

**Figure 13.** Rating curve from numerical simulations (Model #1) and experiments.

Simulations for Model #2 were conducted using three different sizes of the calculation grid (10–20 mm, 8–16 mm, 5–12 mm). For all cases, the simulation continued until the steady state was reached inside the dissipation tank.

As mentioned earlier, velocity values measured with a Pitot tube were used to determine the grid resolution to employ for further analysis. The investigated discharge value was 1450 $m^3$/s and the grid resolutions were those previously defined as low, medium, and high. According to the results obtained for Model #1, the k-omega SST model was used for the simulations.

Figure 14 presents the comparison between the velocity profiles reconstructed with Model #2 at three different resolutions and the Pitot measurements at location 0.14 m upstream the chute, within the surface spillway volume. No remarkable differences can be noted for the different resolutions adopted for the simulations. Conversely, Figure 15 which presents the same comparison at a location of 1.639 m downstream from the chute suggests

that higher resolution is required for a better match. For this reason, further simulations were conducted, employing the higher resolution.

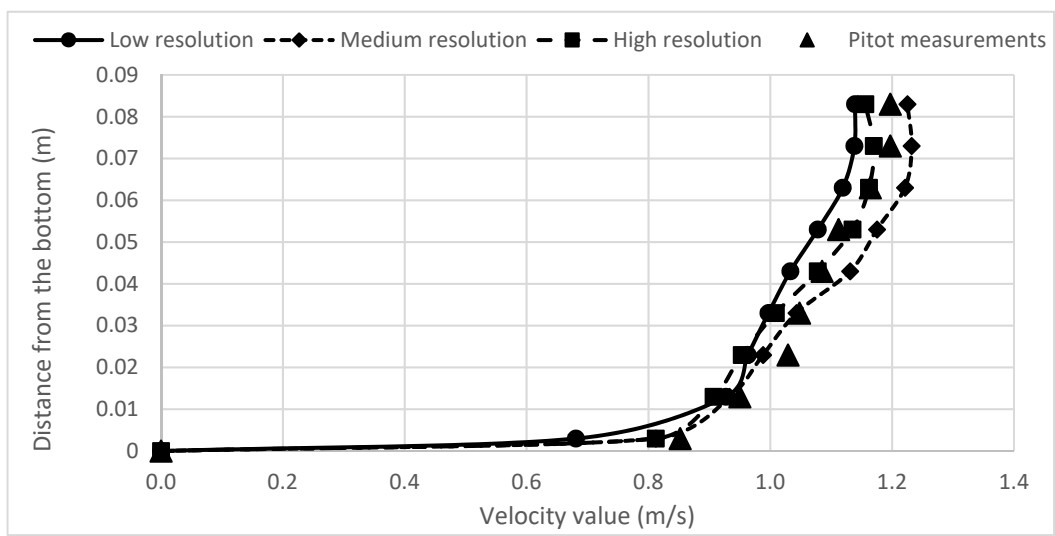

**Figure 14.** Comparison between the velocity profiles reconstructed with Model #2 at three different resolutions and Pitot measurements at location 0.14 m upstream from the chute and within the surface spillway volume.

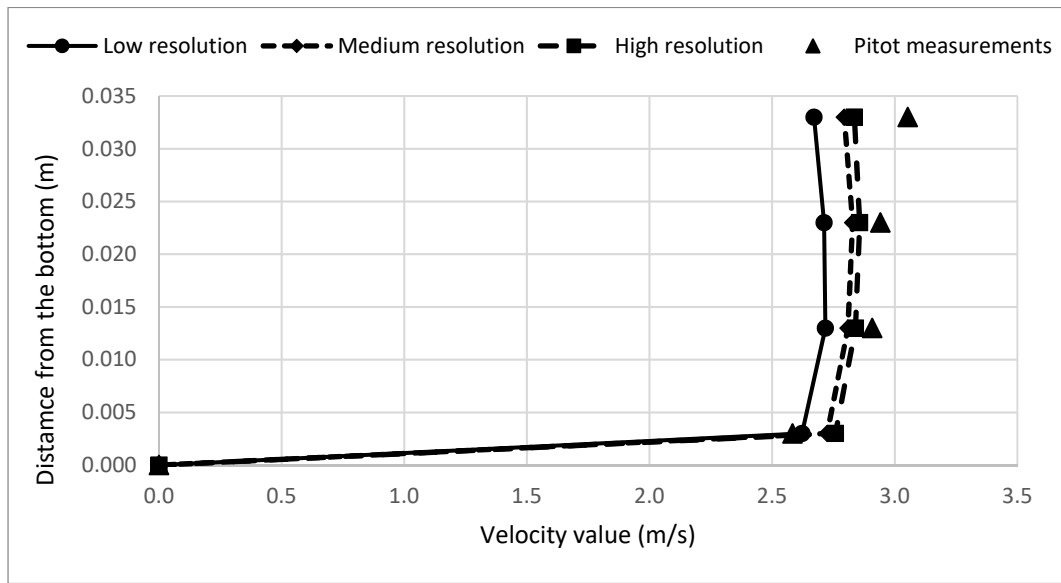

**Figure 15.** Comparison between the velocity profiles reconstructed with Model #2 at three different resolutions and Pitot measurements at location 1.639 m downstream from the chute.

Table 5 presents the details of the simulations conducted recalling that $Q = 830 \text{ m}^3/\text{s}$ was the maximum flow rate discharged from the spillway during the event which occurred in January 2003, characterizing a rainfall event with a return period of 30 years; whereas $Q = 1450 \text{ m}^3/\text{s}$ and $Q = 1650 \text{ m}^3/\text{s}$ correspond to a return period of 100 and 200 years.

**Table 5.** Height of the free surface inside the upstream tank, expected flow rate and the presence of sill #2 for Model #2.

| Height of the Free Surface Inside the Upstream Tank (m) | Expected Discharge (m³/s) | Sill #2 |
| --- | --- | --- |
| 0.24077 | 830 | yes |
| 0.25697 | 1450 | yes |
| 0.25697 | 1450 | no |
| 0.26237 | 1650 | yes |
| 0.26237 | 1650 | no |

Figure 16 presents the water volume fraction reconstructed with Model #2 for a discharge value of 1650 m$^3$/s when sill #2 is included in the numerical model.

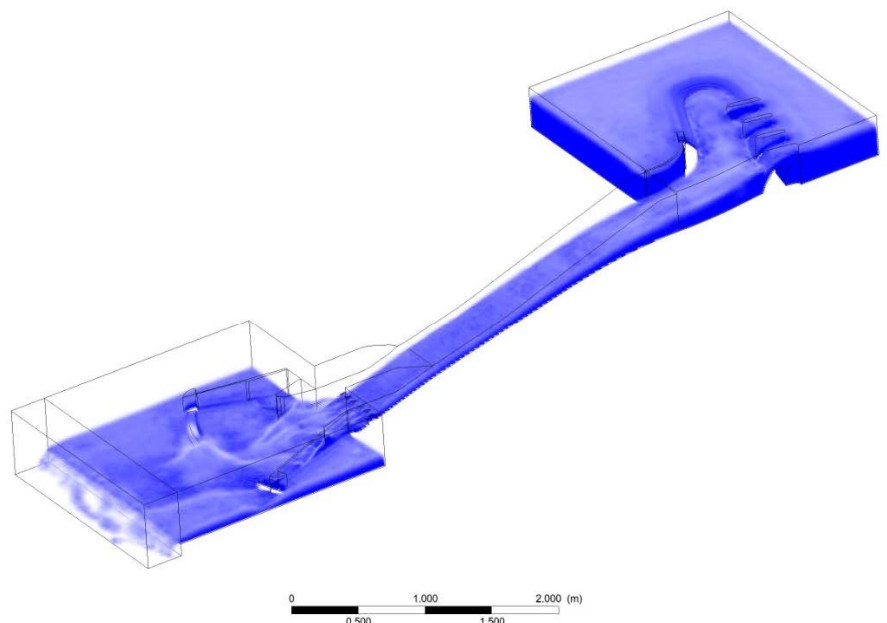

**Figure 16.** Water volume fraction reconstructed with Model #2 for a discharge value of 1650 m$^3$/s and the presence of sill #2.

The effects of sill #2 on the hydrodynamics in the stilling basin and downstream areas are quite evident and may be quantitatively appreciated in Figure 17 where a zoom in the dissipation tank area is presented and compared to the experimental outcomes.

Figure 17 shows the images of the hydrodynamics in the stilling basin and the riverbed downstream from the lateral view. The corresponding images below present the water volume fraction computed by the numerical model. In each experimentally gathered image, a 5 × 5 cm$^2$ mesh (model scale) corresponding to a 3 × 3 m$^2$ mesh at a prototype scale was overlapped over the investigated area. The blue line corresponds to the end of sill #4 while the red line defines the area of impact of the jet outflowing from the spillway chute. The comparison between the images, both the experimental and numerical ones, clearly shows the effect of the ski-jump sill removal, which produces a decrease in the distance between the jet impact area and the stilling basin of roughly 12 m.

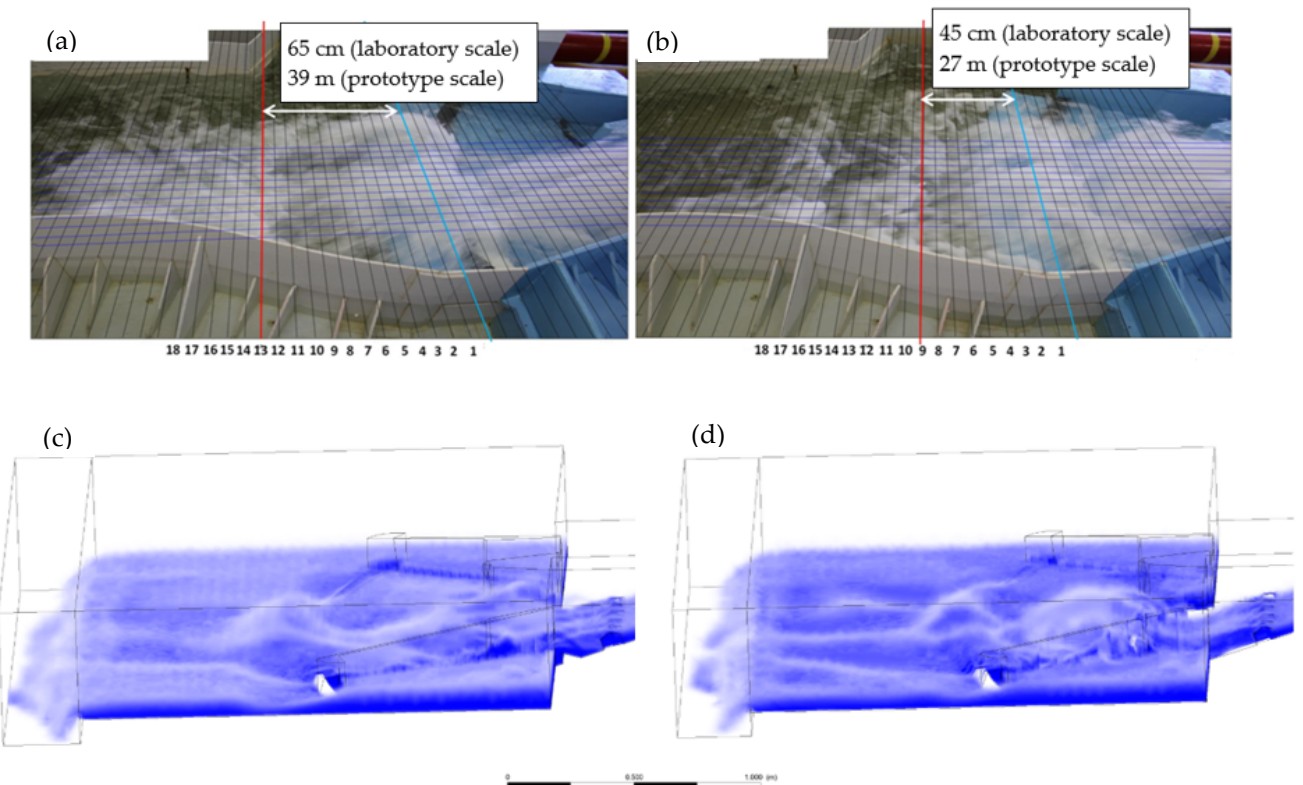

**Figure 17.** Hydrodynamics in the stilling basin and the riverbed downstream from the lateral view for a discharge value of 1650 m³/s and (**a**) ski-jump-like sill in place (**c**) is the output of the numerical model); and (**b**) ski-jump-like sill removed (**d**) is the output of the numerical model).

## 4. Conclusions

In this paper, a comparison between a numerical and experimental model of the Liscione dam was presented. The velocity and free surface elevation were the physical quantities compared. It is not always possible to reproduce a large infrastructure (i.e., a dam) in a laboratory. For this purpose, numerical models can be implemented as a useful alternative. Numerically reproducing a dam also enables it to be made independent of the scale (as it can be reproduced in a prototype scale). Nevertheless, it is mandatory to verify the numerical results with experimental ones to validate the numerical simulations.

In the research presented herein, the results of the numerical simulations confirm the outcomes of the experimental investigation, i.e., the dissipation tank is undersized and therefore insufficient to contain the jet-like flow outflowing from the spillway chute. Due to the high energy content of the current, a further jet-like flow is generated and introduced into the riverbed downstream from the dam with important effects in terms of river bottom erosion. The numerical model also made it possible to compare the hydrodynamics when the ski-jump-like sill is kept or removed from the bottom of the chute. The model clearly shows the beneficial effect achievable with the removal of this sill. It is worth underlining the effectiveness of the commercial software employed in this investigation for the Computational Fluid Dynamics simulations. The numerical model, properly validated with the experimental outcomes, describes the hydrodynamics of the current for the various discharge values under investigation fairly well. Such a validated model can then be employed in the design stage to provide a qualitative visualization of the current for different discharge values and quantitative information on the hydrodynamic features of the flow.

**Author Contributions:** Conceptualization, M.M., M.C. and P.D.G.; methodology, M.M.; validation, M.C., M.M. and P.D.G.; investigation, M.C. and M.M.; data curation, M.C. and M.M.; writing— original draft preparation, M.M.; funding acquisition, P.D.G. All authors have read and agreed to the published version of the manuscript.

**Funding:** This research was partially funded by MoliseAcque.

**Institutional Review Board Statement:** Not applicable.

**Informed Consent Statement:** Not applicable.

**Data Availability Statement:** Data are available upon request.

**Acknowledgments:** The authors wish to thank Molise Acque and Carlo Tatti for funding the present research. The authors also wish to thank Fabio Sammartino for his immeasurable contribution in building the laboratory model; Cosmo Cimorelli and Valerio Ricceri for their invaluable help during the model setup, experimental campaign, and numerical simulations.

**Conflicts of Interest:** The authors declare no conflict of interest. The funders had no role in the design of the study; in the collection, analyses, or interpretation of data; in the writing of the manuscript, or in the decision to publish the results.

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
