# Peer review of "Numerical and Physical Modeling of Ponte Liscione (Guardialfiera, Molise) Dam Spillways and Stilling Basin"

_hydrology, doi:10.3390/hydrology9120214_

Round 1
Reviewer 1 Report
This paper deals with the application of the software 3D Fluent based on the Reynolds Averaged Navier-Stokes equations RANS and the k-ω SST turbulence model in order to simulate the computational fluid dynamics related to various scenario of discharges in the Liscione dam. In particular, two computation domains regarding the surface spillway and the whole dam were constructed. At the same time, a physical model was realized and the numerical results were compared with the observed ones. The topic is very interesting and shows how an appropriate numerical model can be a support to design of the dams considering the effects of climate change. The introduction is well written and the scope of the paper is clear however the results, in my opinion, should be written and shown more clearly.
In my opinion, showing the results in 3D figures is very nice but you lose the perception of the accuracy of the numerical model with respect to the observed results or the differences between the numerical results using different meshes or different scenarios. For this reason I suggest adding graphical comparisons to figures 11, 12, 13,17 and 18 which can be, for example, the profile of the free surface in some interesting sections for the fluid dynamic study
Considering my comments, my recommendation is major Revision.
Author Response
see file attached

Reviewer 2 Report
1- The author should mention their innovation and its behind idea in the abstract and introduction section. Why this research may be useful for other scientist. How can other professional agency can use your work style.
2- Why you choose the ANSYS software. How did you decide that this software is suitable for your work? Did you evaluate other software such as FLOW3D or SSIIM you can also reference to such article in this matter for example., "numerical simulation of flow over rectangular broad crested weir (Real case study)."
3- The text fluency and grammar should improve.
4- I think it is better to manage figure 1,2 and 3 in one figure. All of them show the same content images from the dam. Information about temperature, precipitation, amount of flood in the period of different returns, as well as the floods experienced in the dam and even the characteristics of the catchment area of the dam in the form of these figures can be significant.
5- In the fig3.a, it is clear that green algae have formed under the gate weir. How has the effect of these algae been included in the numerical model?
6- Did you imported the dam bathymetry to you model? How
7- In line 251- 256, please clarify the initial condition. From which source did you set the initial condition.
8- I think it is essential to calibrate your model with the velocity distribution. For this purposes also you should show that your turbulence model also is suitable. Please concentrate on flow filed and velocity distribution in laboratory and it's comparison in the model output.
9- The conclusion section is so general. It is essential to complete this section with your research quantitative output.
Author Response
see file attached

Round 2
Reviewer 1 Report
The authors improved the paper by responding to all recommendations.
Author Response
thanks
Reviewer 2 Report
1- The author reply to comment number 1 is convincing but the author reply is not properly reflected in the text. I think it is better to transfer some part of author reply exactly in the text.
2- The author reply to comment number 2 is convincing and I suggest to add additional table with the ability and advantage (disadvantage) of existing numerical model and highlight the take decision for choosing Fluent.
3- The author reply to comment number 5 is convincing. I think this subject may be proposed by the author for upcoming supplementary research work at the manuscript conclusion section.
4- The author reply to comment number 5 is not convincing. I cannot see the bathymetry in the figures.
Author Response
We wish to thank the reviewer for the constructive and stimulating points they raised which hopefully lead to making our contribution clearer for a wider audience. We believe that we satisfactorily addressed all the issues raised and clarified the Reviewer’s concerns.
In the resubmitted manuscript, changes with respect to the previous version are highlighted using the "Track Changes" function in Microsoft Word.
We include hereafter the point by point response to the Referee. We also include the Referee’s comments.
1- The author reply to comment number 1 is convincing but the author reply is not properly reflected in the text. I think it is better to transfer some part of author reply exactly in the text.
We have included in the final part of the Introduction the reply to the reviewer.
2- The author reply to comment number 2 is convincing and I suggest to add additional table with the ability and advantage (disadvantage) of existing numerical model and highlight the take decision for choosing Fluent.
Unfortunately, we have not the possibility to test the same experiment with other commercial software. For this reason we don't feel like we are skilled enough to make the requested comparison.
3- The author reply to comment number 5 is convincing. I think this subject may be proposed by the author for upcoming supplementary research work at the manuscript conclusion section.
The assumption here is that for good maintenance practices, the dam bottom elements must be free of vegetation or obstacles in general. For this reason, no roughness elements were included in the simulation. We don't feel like this issue should be inserted in the paper because we don't have the expertise in this field, which indeed sounds very interesting.
4- The author reply to comment number 5 is not convincing. I cannot see the bathymetry in the figures.
We have modified the manuscript including a clearer description of the choices concerning the bathymetry. The following paragraph was included in the paper. "Initially, the bathymetry was imported as an STL file built from information gathered from the area Digital Terrain Model. Preliminary tests demonstrated that its influence on the reconstructed water levels was negligible with respect to analogous simulations performed without implemented the lake bottom profile. For this reason, the bathymetry was not included in the final configuration of both Model #1 and #2."